# COVID-19 outbreak–related psychological distress among healthcare trainees: a cross-sectional study in China

Yue Wang,[1] Yuchen Li,[1,2] Jingwen Jiang,[2] Yuying Feng,[3] Donghao Lu ![ORCID] ,[4,5] Wei Zhang ![ORCID] ,[1,2] Huan Song[2,6]

YW and YL contributed equally.

For numbered affiliations see end of article.

**Correspondence to**
Dr Donghao Lu;
donghao.lu@ki.se and
Professor Wei Zhang;
weizhanghx@163.com

## ABSTRACT

**Objectives** The COVID-19 outbreak has caused enormous strain on healthcare systems, and healthcare trainees, which comprise the future healthcare workforce, may be a vulnerable group. It is essential to assess the psychological distress experienced by healthcare trainees during the COVID-19 outbreak.

**Design, setting and participants** A cross-sectional study with 4184 healthcare trainees at Sichuan University in China was implemented during 7–13 February 2020. Participants were grouped by training programmes (medicine, medical technology and nursing) and training stages (undergraduate, postgraduate and residency).

**Main outcomes** COVID-19–related psychological distress and acute stress reaction (ASR) were assessed using the Kessler 6-item Psychological Distress Scale and the Impact of Event Scale–Revised, respectively. We estimated the ORs of distress by comparing trainees across programmes and training stages using multivariable logistic regression.

**Results** Significant psychological distress was found in 1150 (30.90%) participants and probable ASR in 403 (10.74%). Compared with the nursing trainees, the medical trainees (OR 1.54, 95% CI 1.22 to 1.95) reported a higher burden of psychological distress during the outbreak, while the medical technology trainees (OR 1.25, 95% CI 0.97 to 1.62) reported similar symptom scores. Postgraduates (OR 1.55, 95% CI 1.16 to 2.08) in medicine had higher levels of distress than their undergraduate counterparts did, whereas the nursing residents (OR 0.38, 95% CI 0.20 to 0.71) reported a lower burden than did nursing undergraduates. A positive association was found between having active clinical duties during the outbreak and distress (OR 1.17, 95% CI 0.98 to 1.39), particularly among the medical trainees (OR 1.85, 95% CI 1.47 to 2.33) and undergraduates (OR 4.20, 95% CI 1.61 to 11.70). No clear risk patterns of ASR symptoms were observed.

**Conclusions** Medical trainees, particularly postgraduates and those with active clinical duties, were at risk for psychological distress during the COVID-19 outbreak. Stress management may be considered for high-risk healthcare trainees.

## INTRODUCTION

The ongoing global pandemic of the 2019 novel coronavirus disease (COVID-19) has caused 1 991 562 cases and 130 885 deaths

## Strengths and limitations of this study

► We assessed psychological distress among healthcare trainees across different programmes and training stages during the COVID-19 outbreak.
► To shed light on the pandemic's impact on trainees' lives and work, we assessed their concerns and needs during the outbreak and their influence on the future career choices of the trainees without active clinical duties; we also evaluated work–family conflict and support among trainees with clinical duties.
► Our analyses were limited by the study's cross-sectional design, its setting in a single medical school and teaching hospital; hence, the results should be interpreted in light of these limitations and the survey's constraints.

as of 16 April 2020.[1] Witnessing an unexpected illness or death, fear of being in direct contact with and infected by patients with COVID-19, and dealing with household financial hardships during the outbreak have increased the mental burden in the general population.[2] These factors have also elevated the mental burden of healthcare trainees and workers,[3–5] with frontline workers having heavy workloads and being placed at higher risk for COVID-19, due to the drastic surge in patients with COVID-19. Emerging data indicate that Chinese healthcare workers exposed to COVID-19 have experienced psychological symptoms, especially women, nurses, those in Wuhan (the first epicentre) and front-line workers.[6] Other studies have reported a profound mental impact of the COVID-19 outbreak on healthcare workers globally.[3 5 7]

Despite their limited direct contact with patients with COVID-19, healthcare trainees are a vulnerable group.[8] As the pandemic escalates, many countries are considering or have already graduated senior students earlier to assist frontline workers. Other aggressive approaches have been proposed, for instance, suspending all medical school

education for 1 year and recruiting medical students tor testing, tracking and quarantining patients with COVID-19.[9] Although many trainees are inspired during these unprecedented times, some, especially those without sufficient clinical experience, may experience stress. Nevertheless, the psychological state of healthcare trainees across various programmes and training stages, in response to the COVID-19 outbreak, is unknown.

## MATERIALS AND METHODS
### Study design
We conducted a cross-sectional study of healthcare trainees from the West China School of Medicine and West China Hospital, Sichuan University during 7–13 February 2020. We invited 7177 individuals, including 2483 undergraduates, 2606 postgraduates and 2088 residents, to participate in this study to assess their mental health and working conditions during the COVID-19 outbreak via WeChat, a popular social media application in China. The 4184 trainees who agreed to participate were included in the analyses. For data protection, answers to these electronic questionnaires were kept anonymously. The response rates for undergraduates, postgraduates and residents were 73.22%, 71.49% and 24.09%, respectively (online supplemental figure 1).

We focused exclusively on the main concerns and needs of undergraduates and postgraduates who were not involved in clinical work during the COVID-19 epidemic, and the impact of their experiences on their future career plans. We also conducted a short survey of clinical workers about work–family conflict and support during the epidemic. The total number of participants included 1818 undergraduates, 1863 postgraduates and 503 residents.

Most of the undergraduate and postgraduate students were at home throughout the country during the COVID-19 outbreak due to the Chinese Spring Festival, while all residents remained in Chengdu, Sichuan Province because of their clinical duties. As of 6 February 2020, the total number of confirmed COVID-19 cases was 344 in Sichuan Province (102 in Chengdu), and 7226 individuals were under medical observation.[10]

### Healthcare programmes and training stages
Healthcare training programmes in China mainly consist of medicine, medical technology and nursing for the preparation of future doctors, medical technologists (including medical laboratory technologists, imaging technologists, physical therapists and optometrists) and nurses to practice in healthcare settings. The training stages in this study were divided into three categories: undergraduate, postgraduate and residency. All training programmes begin during students' enrolment in undergraduate programmes; the length of training of medical programmes is 5 years, and it is 4 years for medical technology and nursing programmes. After graduation, individuals continue training in a postgraduate programme (3–6 years) with a primary focus on research, which can be combined with clinical training. Students who pursue careers as clinicians enter a residency programme (3 years for medicine and 2 years for medical technology and nursing) for supervised clinical practice after graduation from an undergraduate or postgraduate programme.

Due to the co-occurrence of COVID-19 and the Chinese Spring Festival, individuals in the early stage of training had a low proportion of clinical experiences at the time of the survey. In order to protect students without clinical experience, the medical school cancelled clinical practicums for undergraduates after the COVID-19 outbreak, and a few of the senior undergraduates with internship experiences volunteered to remain at the hospital and support its clinical work. The clinically active trainees included 503 residents and 325 students (304 postgraduates and 21 undergraduates). To assess the work status of the trainees, we asked all participants in the survey, "Are you actively performing clinical duties at this time?" (online supplemental text).

### Assessment of outbreak-related psychological distress
When we assessed psychological distress and acute stress reaction (ASR), we phrased the questions so they were specific to the COVID-19 outbreak (online supplemental text).

Psychological distress was assessed using the Chinese version of the Kessler Psychological Distress Scale (K6). The instrument consists of six items pertaining to major depression and generalised anxiety disorder and asks respondents how frequently they have experienced relevant symptoms during the past month.[11] Each item has five options ranging from 0 (never) to 4 (all of the time), and the total score ranges from 0 to 24. We considered a score ≥5 as clinically significant distress in accordance with the validation studies on Asian populations.[12 13] Cronbach's alpha was 0.91 in our study, indicating good scale reliability.

The variable ASR was evaluated using the Chinese version of the Impact of Event Scale–Revised (ISE-R).[14] The instrument consists of 22 items and yields a total score and scores on the Intrusion, Avoidance and Hyperarousal subscales. Respondents identify a stressful event and how much they were distressed or bothered during the past 7 days by the difficulties listed in the items. Responses are rated on a 5-point scale, ranging from 0 (not at all) to 4 (extremely). Individuals with a score ≥24 points are considered to have probable ASR.[15 16] Cronbach's α was 0.91 in our study, suggesting good scale reliability.

### Assessment of the outbreak's impact
To shed light on the impact of the outbreak on trainees' lives and work, we assessed the concerns and needs of trainees without active clinical duties during the outbreak and the pandemic's influence on their future career choices. We also evaluated work–family conflict and support among the trainees with clinical duties (online supplemental text).

*Concerns and needs during the outbreak, and their influence on future career choices:* To understand trainees' main concerns and needs, we asked the following question. "Under the current circumstances, I am concerned about a) being infected with the novel coronavirus; b) my physical health condition; c) my psychological health; d) academic performance; e) my social life and work; f) my traveling plans; g) the risk of infection from family members or friends; h) my personal and family's financial situation; and i) other issues." We also asked participants to respond to the following item. "If I were to work during the outbreak, I would need: a) personal protective equipment; b) social insurance; c) salary incentives; d) clinical practice guidance; e) professional track record; and f) other needs." Multiple responses were allowed for these questions. We used one single question: "Has the outbreak affected your future career plans?" to assess the impact of the COVID-19 outbreak on trainees' future career plans.

*Work–family conflict and support:* The 9-item Chinese version of the Work–Family Conflict and Support Scale was used to investigate work–family conflict, social support and policy support.[17] Each dimension has three items and each item has three options: 1 (agree), 2 (neutral) and 3 (disagree).

### Statistical analysis

We compared the baseline characteristics of the trainees across the different programmes (ie, medicine, medical technology and nursing) using Student's t-test (for continuous variables) and the $\chi^2$ test (for categorical variables). We described the distributions of the symptoms' scores (transformed z-scores are reported as mean SD), and the proportion of identified cases (corresponding to the cut-off points stated in the Methods section), in each of the three programme groups. Differences in symptom scores or the probability of cases were estimated using linear regression (β coefficients) and logistic regression (ORs), respectively. We examined the associations of the concerns, needs and future career choices with psychological distress and ASR among the participants without active clinical duties, and the associations of family–work conflict with psychological distress and ASR in the participants with active clinical duties. All models were adjusted for age, sex, marital status and epidemic contact characteristics to address confounding by these variables. We also adjusted the model for training programme and training stage when analysing the associations of concerns, needs, career impact and family–work conflicts with psychological distress and ASR. As the status of clinical duty is strongly correlated with training stage, we did not adjust for active clinical duty (yes or no) as covariates. Individuals with missing data on the measures of psychological distress (462, 11.04%) or ASR (433, 10.35%) were not included in the corresponding analyses. We analysed the data anonymously, and all analyses were conducted using R V.3.6.1; p value <0.05 was considered to be statistically significant.

### Patient and public involvement

Patients and/or the public were not involved in the design, or conduct, or reporting, or dissemination plans of this research.

## RESULTS

### Demographic characteristics

The mean age of the 4184 participants was 23.41±3.81 years and 64.89% were females. No confirmed cases of COVID-19 were reported, but 31 (0.74%) trainees had at least one relative who was infected. Approximately one-fifth (19.79%) of participants were involved in active clinical duties; among them, 74 (8.94%) were working on the front lines and 38 (4.59%) had direct contact with patients with COVID-19 (table 1). Compared with trainees in the medical technology and nursing programmes, the trainees in medicine were older (p<0.01) and more likely to be postgraduates (p<0.01), males (p<0.01), married (p<0.01) and living outside of Hubei Province, the first epicentre (p=0.02). At the time of the survey, nursing trainees were more likely to have active clinical duties and work on the front lines (p<0.01).

### Outbreak-related psychological distress and ARS

During the outbreak, 1150 (30.90%) trainees reported significant psychological distress and 403 (10.74%) reported probable ASR. Distress symptoms (β 0.24, 95% CI 0.14 to 0.35) and cases of significant distress (OR 1.54, 95% CI 1.22 to 1.95; table 2) were positively associated with being medical trainees, compared with being nursing trainees. A similar pattern was found among medical technology trainees (β 0.13, 95% CI 0.02 to 0.25; OR 1.25, 95% CI 0.97 to 1.62), although the association with distress cases was not significant. Compared with undergraduates, postgraduates (OR 1.55, 95% CI 1.16 to 2.08) in the medical programme had higher levels of distress, whereas nursing residents (OR 0.38, 95% CI 0.20 to 0.71) reported a lower burden. No significant increase was found across training stages among the medical-technology trainees. Similar patterns, with weaker associations, were observed for symptoms of ASR and probable distress cases across the training programmes and stages within the programmes, except for the lower burden of ASR symptoms reported by the medical residents compared with the undergraduates.

Associations of active clinical duties during the outbreak with distress symptoms (β 0.09, 95% CI 0.01 to 0.18) and cases of significant distress (OR 1.17, 95% CI 0.98 to 1.39; table 3) were found. The association between active duties and distress was positive among the medical trainees (OR 1.85, 95% CI 1.47 to 2.33), but negative among the nursing trainees (OR 0.55, 95% CI 0.32 to 0.93). The association was slightly stronger among undergraduates (OR 4.20, 95% CI 1.61 to 11.70) than it was among postgraduates (OR 2.23, 95% CI 1.72 to 2.91). However, active clinical duty was negatively associated with ASR symptoms (β −0.10, 95% CI −0.19 to −0.02), and except for that finding,

**Table 1** Characteristics of healthcare trainees—N (%) or mean (SD)

| | All | Medicine | Medical technology | Nursing | P value |
|---|---|---|---|---|---|
| Participants, N | 4184 | 2727 (65.18) | 944 (22.56) | 513 (12.26) | |
| Age, years, mean (SD) | 23.41 (3.81) | 24.20 (3.84) | 21.90 (3.24) | 21.98 (3.36) | <0.01 |
| Sex | | | | | <0.01 |
| Male | 1469 (35.11) | 1133 (41.55) | 265 (28.07) | 71 (13.84) | |
| Female | 2715 (64.89) | 1594 (58.45) | 679 (71.93) | 442 (86.16) | |
| Marital status | | | | | <0.01 |
| Married | 331 (7.91) | 254 (9.31) | 48 (5.08) | 29 (5.65) | |
| Unmarried | 3853 (92.09) | 2473 (90.69) | 896 (94.92) | 484 (94.35) | |
| Training stage | | | | | <0.01 |
| Undergraduate | 1791 (42.81) | 940 (34.47) | 588 (62.29) | 263 (51.27) | |
| Postgraduate | 1890 (45.17) | 1662 (60.95) | 142 (15.04) | 86 (16.76) | |
| Residency | 503 (12.02) | 125 (4.58) | 214 (22.67) | 164 (31.97) | |
| Location | | | | | 0.02 |
| Hubei* | 67 (1.60) | 46 (1.69) | 20 (2.12) | 1 (0.19) | |
| Outside Hubei | 4117 (98.40) | 2681 (98.31) | 924 (97.88) | 512 (99.81) | |
| Relatives with COVID-19 | | | | | 0.90 |
| No | 4153 (99.26) | 2706 (99.23) | 937 (99.26) | 510 (99.42) | |
| Yes | 31 (0.74) | 21 (0.77) | 7 (0.74) | 3 (0.58) | |
| Active clinical duty | | | | | <0.01 |
| No | 3356 (80.21) | 2301 (84.38) | 719 (76.17) | 336 (65.50) | |
| Yes | 828 (19.79) | 426 (15.62) | 225 (23.83) | 177 (34.50) | |
| Working position† | | | | | <0.01 |
| Frontline‡ | 74 (8.94) | 36 (8.54) | 5 (2.22) | 33 (18.64) | |
| Second-line | 754 (91.06) | 390 (91.55) | 220 (97.78) | 144 (81.36) | |
| Contact with COVID-19† | | | | | 0.11 |
| Yes | 38 (4.59) | 24 (5.63) | 11 (4.89) | 3 (1.69) | |
| No | 790 (95.41) | 402 (94.37) | 214 (95.11) | 174 (98.31) | |

*Hubei Province was the epicentre at the time of the survey.
†Information was only assessed for participants with activity clinical duty.
‡Frontline working positions was defined as working in departments directly engaging in care for patients with COVID-19, including Emergency, Respiratory, Critical Care Medicine and Infectious Disease Departments.

almost all associations of ASR across training programmes and stages were not significant.

### Outbreak's impact on vulnerable trainees

Among the trainees without active clinical duties during the outbreak, psychological distress was significantly associated with concerns about mental health (OR 2.41, 95% CI 1.90 to 3.04; table 4) and demands for personal protective equipment (OR 1.51, 95% CI 1.07 to 2.16). They were more likely to consider future careers outside of medicine (OR 2.89, 95% CI 1.77 to 4.69). A similar pattern was found for ASR, except for concerns about academic performance, travel plans, personal or family financial hardship, and the need for personal protective equipment, which were only associated with psychological distress.

Among the trainees with active clinical duties, those with distress were more likely to report work–family conflict (ORs 2.20–2.68; table 5). In contrast, adequate social support (ORs 0.42–0.47) and reasonable work arrangements (ORs 0.40–0.47) were associated with lower psychological distress. A similar pattern was found for ASR.

### DISCUSSION

In this large-scale cross-sectional study, we found that psychological distress was common among healthcare trainees during the COVID-19 outbreak. Medical trainees, particularly postgraduates and those with active clinical duties, were at higher risk for psychological distress, compared with those in other training programmes or at

**Table 2** Psychological distress and acute stress reaction among healthcare trainees between different training programmes and stages

| | Psychological distress* | | | | Acute stress reaction† | | | |
| --- | --- | --- | --- | --- | --- | --- | --- | --- |
| | Symptoms (z-score) | | Cases | | Symptoms (z-score) | | Cases (403) | |
| | Mean±SD | β (95% CI)‡ | N (%) | OR (95% CI)‡ | Mean±SD | β (95% CI)‡ | N (%) | OR (95% CI)‡ |
| **Training programme** | | | | | | | | |
| Medicine | 0.06±1.05 | 0.24 (0.14 to 0.35) | 792 (33.22) | 1.54 (1.22 to 1.95) | 0.01±1.01 | 0.14 (0.04 to 0.24) | 256 (10.66) | 1.08 (0.78 to 1.52) |
| Medical technology | −0.07±0.92 | 0.13 (0.02 to 0.25) | 242 (27.94) | 1.25 (0.97 to 1.62) | 0.03±0.99 | 0.13 (0.02 to 0.24) | 96 (11.00) | 1.06 (0.74 to 1.54) |
| Nursing | −0.18±0.85 | Ref. | 116 (24.58) | Ref. | −0.08±0.95 | Ref. | 51 (10.71) | Ref. |
| **Medicine** | | | | | | | | |
| **By training stage** | | | | | | | | |
| Undergraduate | −0.10±0.99 | Ref. | 223 (26.05) | Ref. | −0.01±1.06 | Ref. | 86 (9.92) | Ref. |
| Postgraduate | 0.16±1.08 | 0.23 (0.09 to 0.37) | 529 (37.23) | 1.55 (1.16 to 2.08) | 0.02±0.99 | 0.05 (−0.08 to 0.19) | 163 (11.39) | 1.14 (0.74 to 1.78) |
| Residency | 0.10±0.90 | 0.14 (−0.09 to 0.37) | 40 (37.38) | 1.45 (0.90 to 2.31) | −0.07±0.84 | −0.08 (−0.31 to 0.15) | 7 (6.73) | 0.58 (0.22 to 1.31) |
| **Medical technology** | | | | | | | | |
| **By training stage** | | | | | | | | |
| Undergraduate | −0.07±0.97 | Ref. | 155 (28.23) | Ref. | 0.06±1.05 | Ref. | 66 (11.98) | Ref. |
| Postgraduate | −0.02±0.82 | −0.14 (−0.40 to 0.12) | 36 (30.00) | 0.83 (0.44 to 1.55) | 0.01±0.90 | −0.09 (−0.37 to 0.19) | 11 (8.80) | 0.62 (0.23 to 1.56) |
| Residency | −0.10±0.84 | −0.15 (−0.35 to 0.04) | 51 (25.89) | 0.73 (0.45 to 1.17) | −0.06±0.86 | −0.15 (−0.36 to 0.06) | 19 (9.64) | 0.72 (0.36 to 1.43) |
| **Nursing** | | | | | | | | |
| **By training stage** | | | | | | | | |
| Undergraduate | −0.10±0.90 | Ref. | 75 (29.88) | Ref. | 0.05±1.05 | Ref. | 32 (12.75) | Ref. |
| Postgraduate | −0.02±0.96 | 0.06 (−0.29 to 0.41) | 21 (29.58) | 0.92 (0.35 to 2.30) | 0.04±0.88 | −0.04 (−0.42 to 0.33) | 8 (10.67) | 0.77 (0.18 to 2.81) |
| Residency | −0.39±0.67 | −0.25 (−0.46 to −0.05) | 20 (13.33) | 0.38 (0.20 to 0.71) | −0.36±0.73 | −0.39 (−0.62 to −0.16) | 11 (7.33) | 0.55 (0.23 to 1.28) |

*In this analysis, 462 (11.04%) individuals who missed the measure of psychological distress were not included.
†In this analysis, 433 (10.35%) individuals who missed the measure of acute stress reaction were not included.
‡Estimates were adjusted for age, sex (male or female), marital status (married or unmarried), location (Hubei or outside Hubei) and relatives with COVID-19 (yes or no).

**Table 3** Psychological distress and acute stress reaction among healthcare trainees with and without active clinical duty

| | Psychological distress* | | | | Acute stress reaction† | | | |
| | Symptoms (z-score) | | Cases | | Symptoms (z-score) | | Cases | |
| | Mean±SD | β (95% CI)‡ | N (%) | OR (95% CI)‡ | Mean±SD | β (95% CI)‡ | N (%) | OR (95% CI)‡ |
|---|---|---|---|---|---|---|---|---|
| **Active clinical duty** | | | | | | | | |
| No | −0.03±0.99 | Ref. | 882 (29.73) | Ref. | 0.02±1.02 | Ref. | 326 (10.84) | Ref. |
| Yes | 0.12±1.02 | 0.09 (0.01 to 0.18) | 268 (35.50) | 1.17 (0.98 to 1.39) | −0.07±0.92 | −0.10 (−0.19 to −0.02) | 77 (10.36) | 0.93 (0.71 to 1.22) |
| **By training programme** | | | | | | | | |
| **Medicine** | | | | | | | | |
| No duty | −0.00§±1.02 | Ref. | 609 (30.53) | Ref. | 0.01±1.02 | Ref. | 212 (10.47) | Ref. |
| With duty | 0.39±1.10 | 0.36 (0.24 to 0.47) | 183 (47.04) | 1.85 (1.47 to 2.33) | −0.008§±0.98 | −0.01 (−0.12 to 0.11) | 44 (11.67) | 1.12 (0.77 to 1.58) |
| **Medical technology** | | | | | | | | |
| No duty | −0.08±0.94 | Ref. | 184 (27.84) | Ref. | 0.04±1.02 | Ref. | 76 (11.38) | Ref. |
| With duty | −0.06±0.86 | −0.04 (−0.20 to 0.12) | 58 (28.29) | 0.94 (0.64 to 1.38) | −0.03±0.88 | −0.08 (−0.25 to 0.09) | 20 (9.76) | 0.89 (0.50 to 1.57) |
| **Nursing** | | | | | | | | |
| No duty | −0.10±0.90 | Ref. | 89 (28.62) | Ref. | 0.03±1.01 | Ref. | 38 (12.06) | Ref. |
| With duty | −0.33±0.75 | −0.19 (−0.37 to −0.01) | 27 (16.77) | 0.55 (0.32 to 0.93) | −0.30±0.78 | −0.30 (−0.49 to −0.10) | 13 (8.07) | 0.69 (0.32 to 1.40) |
| **By training stage** | | | | | | | | |
| **Undergraduate** | | | | | | | | |
| No duty | −0.10±0.96 | Ref. | 442 (26.98) | Ref. | 0.02±1.05 | Ref. | 181 (10.96) | Ref. |
| With duty | 0.54±1.18 | 0.62 (0.17 to 1.08) | 11 (61.11) | 4.20 (1.61 to 11.70) | −0.05±1.16 | 0.01 (−0.49 to 0.51) | 3 (16.67) | 1.75 (0.39 to 5.58) |
| **Postgraduate** | | | | | | | | |
| No duty | 0.06±1.02 | Ref. | 440 (33.11) | Ref. | 0.01±0.97 | Ref. | 145 (10.69) | Ref. |
| With duty | 0.52±1.14 | 0.49 (0.35 to 0.62) | 146 (51.59) | 2.23 (1.72 to 2.91) | 0.06±1.03 | 0.07 (−0.06 to 0.20) | 37 (13.50) | 1.35 (0.90 to 1.97) |
| **Residency** | | | | | | | | |
| No duty | – | Ref. | – | Ref. | – | Ref. | – | Ref. |
| With duty | −0.15±0.82 | – | 111 (24.45) | – | −0.16±0.82 | – | 37 (8.20) | – |

*In this analysis, 462 (11.04%) individuals who missed the measure of psychological distress were not included.
†In this analysis, 433 (10.35%) individuals who missed the measure of acute stress reaction were not included.
‡Estimates were adjusted for age, sex (male or female), marital status (married or unmarried), location (Hubei or outside Hubei) and relatives with COVID-19 (yes or no).
§−0.00: <−0.01.

**Table 4** Associations of concerns and needs during COVID-19 outbreaks with psychological distress and acute stress reaction among healthcare trainees without active clinical duty

| | Psychological distress* | | | Acute stress reaction† | | |
|---|---|---|---|---|---|---|
| | No (n=2085) | Yes (n=882) | | No (n=2682) | Yes (n=326) | |
| | N (%) | N (%) | OR (95% CI)‡ | N (%) | N (%) | OR (95% CI)‡ |
| **Concerns** | | | | | | |
| Being infected with the novel coronavirus | 1298 (66.84) | 644 (33.16) | 1.33 (1.09 to 1.61) | 1713 (87.00) | 256 (13.00) | 1.42 (1.05 to 1.95) |
| Physical health condition | 458 (59.17) | 316 (40.83) | 1.41 (1.16 to 1.72) | 630 (80.36) | 154 (19.64) | 1.81 (1.37 to 2.39) |
| Psychological health | 189 (46.78) | 215 (53.22) | 2.41 (1.90 to 3.04) | 294 (71.01) | 120 (28.99) | 3.24 (2.42 to 4.31) |
| Academic performance | 1195 (66.24) | 609 (33.76) | 1.30 (1.09 to 1.56) | 1606 (87.95) | 220 (12.05) | 0.97 (0.74 to 1.27) |
| Social life and work | 383 (64.37) | 212 (35.63) | 1.19 (0.95 to 1.48) | 505 (83.61) | 99 (16.39) | 1.38 (1.01 to 1.87) |
| Travelling plans | 141 (58.02) | 102 (41.98) | 1.36 (1.01 to 1.82) | 201 (81.38) | 46 (18.62) | 1.22 0.81 to 1.79) |
| Family members or friends being infected with the novel coronavirus | 1297 (68.41) | 599 (31.59) | 0.94 (0.78 to 1.13) | 1688 (87.96) | 231 (12.04) | 0.91 (0.69 to 1.21) |
| Personal and family's financial situation | 249 (59.29) | 171 (40.71) | 1.27 (1.01 to 1.61) | 353 (82.48) | 75 (17.52) | 1.18 (0.85 to 1.61) |
| **Needs** | | | | | | |
| Personal protective equipment | 1907 (69.50) | 837 (30.50) | 1.51 (1.07 to 2.16) | 2485 (89.26) | 299 (10.74) | 0.74 (0.48 to 1.18) |
| Social insurance | 1819 (69.24) | 808 (30.76) | 1.31 (0.98 to 1.76) | 2371 (89.03) | 292 (10.97) | 0.86 (0.58 to 1.32) |
| Salary incentives | 1386 (68.21) | 646 (31.79) | 1.37 (1.12 to 1.67) | 1809 (87.69) | 254 (12.31) | 1.79 (1.32 to 2.44) |
| Clinical practice guidance | 1317 (70.58) | 549 (29.42) | 0.89 (0.73 to 1.08) | 1672 (88.42) | 219 (11.58) | 1.13 (0.85 to 1.50) |
| Professional track record | 1213 (70.16) | 516 (29.84) | 0.91 (0.75 to 1.11) | 1551 (88.63) | 199 (11.37) | 0.92 (0.69 to 1.23) |
| **Future career choice** | | | | | | |
| Healthcare worker | 1256 (73.80) | 446 (26.20) | Ref. | 1587 (91.89) | 140 (8.11) | Ref. |
| Medicine related—but not bedside | 618 (69.52) | 271 (30.84) | 1.26 (1.05 to 1.52) | 791 (87.89) | 109 (12.11) | 1.54 (1.17 to 2.02) |
| Outside of medicine | 36 (50.70) | 35 (49.30) | 2.89 (1.77 to 4.69) | 51 (72.86) | 19 (27.14) | 4.10 (2.29 to 7.09) |
| Indeterminate | 175 (57.38) | 130 (42.62) | 2.11 (1.64 to 2.72) | 253 (81.35) | 58 (18.65) | 2.57 (1.82 to 3.58) |

*In this analysis, 389 (11.59%) individuals who missed the measure of psychological distress were not included.
†In this analysis, 348 (10.37%) individuals who missed the measure of acute stress reaction were not included.
‡Estimates were adjusted for age, sex (male or female), marital status (married or unmarried), location (Hubei or outside Hubei), relatives with COVID-19 (yes or no), training programmes (medicine, medical technology or nursing) and training stage (undergraduate, postgraduate or residency).

an earlier training stage. Concerns about mental health were strongly correlated with psychological distress among trainees with no clinical duties, whereas work–family conflict was the greatest concern of distressed trainees with active clinical duties.

The strain of COVID-19 on healthcare systems, medical trainees and other practitioners is challenging. Consistent with prior research,[6 18 19] our results indicated that psychological distress was common among healthcare trainees, especially those in medical training programmes. Conversely, several studies reported that nurses providing care for confirmed or suspected COVID-19 cases had a greater mental burden than doctors.[6 20] Given the small proportion of participants (4.59%) who had direct contact with infected patients, these inconsistent estimates do not invalidate each other.

Another important finding was the association of being in the advanced training stage and having active clinical duties with higher levels of psychological distress. Academic pressures, workload and financial burden

increase with level of training, which could, consequently, increase the mental vulnerability of senior trainees and ultimately contribute to negative mental outcomes.[21–25] During the COVID-19 outbreak, senior medical students and residents were encouraged to assist hospital staff with clinical work to deal with the severe workforce shortage.[26 27] The overwhelming workload and high risk of exposure to COVID-19 might have added to their mental burden. We consistently observed a higher risk of psychological distress across all training stages among individuals who were involved in active clinical duties during the outbreak.

ASR often develops following direct exposure to traumatic events, such as experiencing the COVID-19 outbreak in the epicentre and being charged with the direct care (diagnosis and treatment) of patients confirmed with COVID-19.[6 28] However, few participants were quarantined in the epicentre or had direct contact with infected patients at the time of the survey. The risk for ASR was relatively low and differences between the

**Table 5** Associations of family–work conflicts during COVID-19 outbreak with psychological distress and acute stress reaction among healthcare trainees with active clinical duty

| | Psychological distress* | | | Acute stress reaction† | | |
|---|---|---|---|---|---|---|
| | No (n=487) | Yes (n=268) | | No (n=666) | Yes (n=77) | |
| | N (%) | N (%) | OR (95% CI)‡ | N (%) | N (%) | OR (95% CI)‡ |
| **Work–family conflict** | | | | | | |
| Difficult to care for family due to work | | | | | | |
| Agree | 122 (18.32) | 28 (36.36) | 2.53 (1.67 to 3.84) | 59 (12.11) | 93 (34.70) | 1.86 (1.06 to 3.22) |
| Neutral | 331 (49.70) | 42 (54.55) | Ref. | 249 (5.13) | 135 (50.37) | Ref. |
| Disagree | 213 (31.98) | 7 (9.09) | 0.47 (0.31 to 0.72) | 179 (36.76) | 40 (14.93) | 0.25 (0.10 to 0.55) |
| Family responsibilities affected work | | | | | | |
| Agree | 37 (5.56) | 14 (18.18) | 2.20 (1.16 to 4.29) | 19 (3.90) | 33 (12.31) | 2.56 (1.22 to 5.20) |
| Neutral | 265 (39.79) | 44 (57.14) | Ref. | 172 (35.32) | 145 (54.10) | Ref. |
| Disagree | 364 (54.65) | 19 (24.88) | 0.39 (0.28 to 0.56) | 296 (60.78) | 90 (33.58) | 0.31 (0.17 to 0.54) |
| Difficulties in juggling work and family | | | | | | |
| Agree | 36 (5.41) | 14 (18.18) | 2.68 (1.35 to 5.56) | 14 (2.87) | 37 (13.81) | 2.44 (1.13 to 5.12) |
| Neutral | 246 (36.94) | 43 (55.84) | Ref. | 155 (31.83) | 140 (52.24) | Ref. |
| Disagree | 384 (57.66) | 20 (25.97) | 0.37 (0.26 to 0.53) | 318 (65.30) | 91 (33.96) | 0.28 (0.15 to 0.50) |
| **Social support** | | | | | | |
| Support from family | | | | | | |
| Agree | 466 (69.97) | 35 (45.45) | 0.47 (0.33 to 0.67) | 359 (73.72) | 148 (55.22) | 0.41 (0.24 to 0.71) |
| Neutral | 166 (24.92) | 31 (40.26) | Ref. | 107 (21.97) | 95 (35.45) | Ref. |
| Disagree | 34 (5.11) | 11 (14.29) | 1.16 (0.59 to 2.33) | 21 (4.31) | 25 (9.33) | 1.62 (0.70 to 3.55) |
| Support from colleagues | | | | | | |
| Agree | 545 (81.83) | 44 (57.14) | 0.42 (0.28 to 0.63) | 414 (85.01) | 181 (67.54) | 0.32 (0.19 to 0.55) |
| Neutral | 110 (16.52) | 30 (38.96) | Ref. | 65 (13.35) | 81 (30.22) | Ref. |
| Disagree | 11 (1.65) | 3 (3.90) | 0.51 (0.15 to 1.70) | 8 (1.64) | 6 (2.24) | 0.99 (0.21 to 3.57) |
| Support from supervisors | | | | | | |
| Agree | 505 (75.83) | 30 (38.96) | 0.42 (0.29 to 0.60) | 389 (79.98) | 150 (55.97) | 0.22 (0.13 to 0.38) |
| Neutral | 146 (21.92) | 39 (50.65) | Ref. | 87 (17.86) | 105 (39.18) | Ref. |
| Disagree | 15 (2.25) | 8 (10.39) | 0.86 (0.35 to 2.14) | 11 (2.26) | 13 (4.85) | 2.32 (0.85 to 6.04) |
| **Policy support** | | | | | | |
| Reasonable holiday arrangement | | | | | | |
| Agree | 463 (69.52) | 28 (36.36) | 0.44 (0.30 to 0.65) | 369 (75.77) | 126 (47.01) | 0.27 (0.15 to 0.48) |
| Neutral | 156 (23.42) | 34 (44.16) | Ref. | 94 (19.30) | 102 (38.06) | Ref. |
| Disagree | 47 (7.06) | 15 (19.48) | 1.32 (0.72 to 2.44) | 24 (4.93) | 40 (14.93) | 1.51 (0.72 to 3.07) |
| Reasonable duty arrangement | | | | | | |
| Agree | 468 (70.27) | 29 (37.66) | 0.47 (0.32 to 0.68) | 371 (76.18) | 129 (48.13) | 0.27 (0.15 to 0.47) |
| Neutral | 163 (24.47) | 37 (48.05) | Ref. | 100 (20.53) | 106 (39.55) | Ref. |
| Disagree | 35 (5.26) | 11 (14.29) | 1.69 (0.86 to 3.40) | 16 (3.29) | 33 (12.31) | 1.27 (0.56 to 2.73) |
| Flexible policies to balance family and work | | | | | | |
| Agree | 438 (65.77) | 25 (32.47) | 0.40 (0.28 to 0.58) | 356 (73.10) | 109 (40.67) | 0.29 (0.16 to 0.52) |
| Neutral | 200 (30.03) | 36 (46.75) | Ref. | 120 (24.64) | 123 (45.90) | Ref. |
| Disagree | 28 (4.20) | 16 (20.78) | 3.09 (1.52 to 6.74) | 11 (2.26) | 36 (13.43) | 3.47 (1.63 to 7.24) |

*In this analysis, 73 (8.82%) individuals who missed the measure of psychological distress were not included.
†In this analysis, 85 (10.27%) individuals who missed the measure of acute stress reaction were not included.
‡Estimates were adjusted for age, sex (male or female), marital status (married or unmarried), relatives with COVID-19 (yes or no), working position (frontline or second-line), contact with COVID-19 (yes or no), training programmes (medicine, medical technology or nursing) and training stage (undergraduate, postgraduate or residency).

training programmes and stages were small. Interestingly, we found that trainees with active clinical duties had fewer ASR symptoms during the outbreak.

Our findings suggest that being infected by COVID-19 was the healthcare trainees' leading concern, followed by concerns about their mental health, with regards to psychological distress and ASR. Adequate personal protective equipment and salary incentives might help reduce psychological distress, which is consistent with the finding that family-income stability is a protective factor against anxiety among medical students.[29] Among clinical workers, work–family conflict was positively associated with psychological distress and negatively associated with social support. Therefore, adequate personal protection, timely psychological interventions, a stable financial situation, a strong family and social support may be key factors in reducing the risk of psychological distress among healthcare trainees. Competent leadership, including active participation in outbreak preparedness and making reasonable work arrangements, could also alleviate the emotional strain on healthcare trainees, suggesting the importance of work polices for healthcare trainees.

Studies have found that experiencing psychological distress during the training stage leads to changes in career paths.[30–32] These results are consistent with our finding that healthcare trainees who decided to work in non-medical fields in the future tended to have higher levels of psychological distress, compared with the trainees determined to continue on their original paths. During the COVID-19 outbreak, healthcare trainees might have been emotionally vulnerable to the crisis, because of being knowledgeable about medicine, which increased their awareness of the dangers during the outbreak's early stages. The epidemic represents an extreme situation in which being a 'doctor' is considered a demanding job with social responsibilities, which might have scared or inspired medical trainees, especially those without clinical experience. It is therefore, possible that this crisis also influenced their career choices.

## Limitations

Our study has several limitations. First, given the nature of cross-sectional analyses, our data do not indicate changes in psychological distress from the pre-pandemic period; rather, they characterise the burden during the COVID-19 outbreak. Second, the response rate was low among the residents, and those who did not participate might have been the trainees with highest stress levels at work. The participation rates of undergraduate and postgraduate trainees were satisfactory (73.22% and 71.49%, respectively). Such selection is not likely to provide a thorough explanation of our findings. Third, we only measured distress symptoms once in the early phase of the outbreak. Longitudinal studies are needed in the future, as symptoms may change over time. Fourth, although the effects of age, sex, training programme and training stage were adjusted for their corresponding data

analyses, residual confounding remains because data on other confounding factors were inapplicable (ie, marital status, current location, job position) or not collected (ie, socioeconomic status). Last, although the trainees came from all parts of China, our study was conducted at a single medical school and teaching hospital. The generalisability of our findings to other hospitals and medical populations remains unclear and, therefore, needs further investigation.

## Conclusions

Our findings suggest that psychological distress in response to the COVID-19 outbreak is common among healthcare trainees in China. Medical trainees, particularly postgraduates and with active clinical duties, were at higher risk for psychological distress than the other groups of trainees. Stress management should be provided for high-risk healthcare trainees during the outbreak, particularly if or when the training is accelerated, and trainees join the front lines of the workforce.

**Author affiliations**
[1]Mental Health Center, West China Hospital, Sichuan University, Chengdu, China
[2]West China Biomedical Big Data Center, West China Hospital, Sichuan University, Chengdu, China
[3]Student Affairs Office, West China Hospital, Sichuan University, Chengdu, China
[4]Clinical Research Center for Breast Diseases, West China Hospital, Sichuan University, Chengdu, China
[5]Department of Medical Epidemiology and Biostatistics, Karolinska Institutet, Stockholm, Sweden
[6]Center of Public Health Sciences, University of Iceland, Reykjavik, Iceland

**Acknowledgements** The authors thank Lie Zhang (West China School of Medicine of Sichuan University, Chengdu, China) for coordinating the data collection; Fenfen Ge (West China Hospital of Sichuan University, Chengdu, China), Ting Liu (West China Hospital of Sichuan University, Chengdu, China) and Xiao Liao (Southwest University, Chongqing, China) for data collection. Mr Zhang, Miss Ge, Miss Liu and Miss Liao have no conflicts of interest to declare. We also thank the participating students at the West China School of Medicine.

**Contributors** WZ, DL and HS had full access to all of the data in the study and take responsibility for the integrity of the data and the accuracy of the data analysis. YW and YL contributed equally to the work. Concept and design: YW, YL, WZ. Acquisition, analysis or interpretation of data: YW, YL, HS, YF. Drafting of the manuscript: YW, YL, HS, DL. Critical revision of the manuscript for important intellectual content: all authors. Statistical analysis: YW, JJ. Obtained funding: WZ, HS, DL. Administrative, technical or material support: WZ, HS, DL, YF. Supervision: WZ, HS, DL. All authors have contributed significantly to this work and have met the qualification of authorship.

**Funding** This research was supported by the National Natural Science Foundation of China (No. 81971262 to HS, No. 81801359 to DL), Swedish Research Council (No. 2018-00648 to DL), West China Hospital COVID-19 Epidemic Science and Technology Project (No. HX-2019-nCoV-014 to HS, No. HX-2019-nCoV-019 to WZ) and Sichuan University Emergency Grant (No. 2020scunCoVyingji1002 to HS, No. 2020scunCoVyingji1005 to DL).

**Competing interests** None declared.

**Patient consent for publication** Not required.

**Ethics approval** This study was approved by the Ethics Committee of the Sichuan University, and electronic consent forms were obtained from all participants.

**Provenance and peer review** Not commissioned; externally peer reviewed.

**Data availability statement** The data that support the findings of this study are available from the corresponding author, WZ, on reasonable request.

**ORCID iDs**
Donghao Lu http://orcid.org/0000-0002-4186-8661
Wei Zhang http://orcid.org/0000-0003-3113-9577

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
