## [Reviewer comments · BMJ Open]

ARTICLE DETAILS

TITLE (PROVISIONAL)	COVID-19 outbreak-related psychological distress among healthcare trainees: a cross-sectional study in China
AUTHORS	Wang, Yue; Li, Yuchen; Jiang, Jingwen; Feng, Yuying; Song, Huan; Zhang, Wei; Lu, Donghao

VERSION 1 – REVIEW

REVIEWER	Dr. rer. medic. Moritz Bruno Petzold Charité Universitätsmedizin Berlin Department for Psychiatry and Psychotherapy Charité Campus Mitte Berlin, Germany
REVIEW RETURNED	10-Jul-2020

GENERAL COMMENTS	Thank you very much for the opportunity to review this manuscript. The authors present a cross-sectional examination of psychological burden in medical trainees in a single medical hospital in Sichuan, China. The manuscript addresses a very important topic, where substantial research is urgently needed. The paper is in large parts well written, concise, informative and substantial. The authors recruited a fairly large sample. Despite some limitations (single hospital, cross-sectional), the paper does come with substantial information and does contribute something to this important and timely area of research. Nevertheless, there are some points which would have to be addressed before the manuscript could be published: General points: - The manuscript needs a careful re-read to correct some typographical and grammatical errors. E.g. Abstract: „To assess psychological distress among health professional trainees during the COVID-19 outbreak are necessary, P10,L38: „liner“, P11,L12: „then“ etc- You are changing between two and three decimal places, I think it would be good to unify this. I think two decimal places are sufficient. Abstract: - P3,L42: „No evident increase was found in trainees of medical technology.“ The use of the word increase suggests that a development over time would have been assessed, which was not the case. I would suggest to change wording here.- P3, L47: „(ORs 1.62-1.66)“ Is this the confidence interval? The OR estimate itself seems missing (compare with the next bracket where you report a OR followed by the corresponding CI)- P3,L47: „whereas a lower burden endorsed by nursing residents (OR 0.35, 95% CI 0.19-0.63)“ Compared to whom? The beginning
---

of the sentence says „compared to undergraduate students“. I think this cant be correct as this are two different levels of comparison (training stage vs. training program)

Strenghts and limitations:

- P5,L6: I would suggest to erase the word „comprehensively“. In my opinion you used well suited and realiable instruments to assess psychological distress, nevertheless, psychological distress is a very broad concept and a „comprehensive“ assessment would habe to include a more detailed assessment in my opinion.

Introduction:

- The introduction is very short. I think it would be nescessary to introduce more literature on specific stressors in health professionals and to refer to the multiple commentaries and studies that got published in the last weeks. A short review of the current body of research, what kind of studies exist, what are the first insights and what are methodological problems (in brief) would be important in my opinion. To help you with that here some papers that might be suited for that. This is meant as help and I dont consider it nescessary that you cite all and exactly these paper.

o Daugherty, A. M., & Arble, E. P. (2020). Prevalence of mental health symptoms in residential healthcare workers in Michigan during the covid-19 pandemic. *Psychiatry Research*, 291, 113266. <https://doi.org/10.1016/j.psychres.2020.113266>

o Chew, N. W. S., Lee, G. K. H., Tan, B. Y. Q., Jing, M., Goh, Y., Ngiam, N. J. H., . . . Sharma, V. K. (2020). A multinational, multicentre study on the psychological outcomes and associated physical symptoms amongst healthcare workers during COVID-19 outbreak. *Brain, Behavior, and Immunity*. Advance online publication. <https://doi.org/10.1016/j.bbi.2020.04.049>

o Naser, A. Y., Dahmash, E. Z., Al-Rousan, R., Alwafi, H., Alrawashdeh, H. M., Ghoul, I., . . . Alyami, H. S. (2020). Mental health status of the general population, healthcare professionals, and university students during 2019 coronavirus disease outbreak in Jordan: A cross-sectional study. *Brain and Behavior*, e01730. <https://doi.org/10.1002/brb3.1730>

o Shechter, A., Diaz, F., Moise, N., Anstey, D. E., Ye, S., Agarwal, S., . . . Abdalla, M. (2020). Psychological distress, coping behaviors, and preferences for support among New York healthcare workers during the COVID-19 pandemic. *General Hospital Psychiatry*, 66, 1–8. <https://doi.org/10.1016/j.genhosppsy.2020.06.007>

o Tan, B. Y. Q., Chew, N. W. S., Lee, G. K. H., Jing, M., Goh, Y., Yeo, L. L. L., . . . Sharma, V. K. (2020). Psychological Impact of the COVID-19 Pandemic on Health Care Workers in Singapore. *Annals of Internal Medicine*. Advance online publication. <https://doi.org/10.7326/M20-1083>

o Zerbini, G., Ebigbo, A., Reicherts, P., Kunz, M., & Messman, H. (2020). Psychosocial burden of healthcare professionals in times of COVID-19 - a survey conducted at the University Hospital Augsburg. *German Medical Science : GMS E-Journal*, 18, Doc05. <https://doi.org/10.3205/000281>

o Petzold, M. B., Plag, J., & Ströhle, A. (2020). Dealing with psychological distress by healthcare professionals during the COVID-19 pandemia [Dealing with psychological distress by healthcare professionals during the COVID-19 pandemia]. *Der Nervenarzt*, 91(5), 417–421. <https://doi.org/10.1007/s00115-020-00905-0>

- Furthermore, some introducing sentences on the impact of the pandemic on the general population and that health professionals of course also suffer from the general factors (and in addition to specific factors related to their profession) would be interesting for the reader I think. Again some help with some papers which might be suitable (but again: I do not expect to cite all of them):

- o Le Shi, Lu, Z.-A., Que, J.-Y., Huang, X.-L., Liu, L., Ran, M.-S., . . . Lu, L. (2020). Prevalence of and Risk Factors Associated With Mental Health Symptoms Among the General Population in China During the Coronavirus Disease 2019 Pandemic. *JAMA Network Open*, 3(7), e2014053. <https://doi.org/10.1001/jamanetworkopen.2020.14053>
- o Petzold, M. B., Bendau, A., Plag, J., Pyrkosch, L., Mascarell Maricic, L., Betzler, F., . . . Ströhle, A. (2020). Risk, resilience, psychological distress, and anxiety at the beginning of the COVID-19 pandemic in Germany. *Brain and Behavior*, e01745. <https://doi.org/10.1002/brb3.1745>
- o Ueda, M., Stickley, A., Sueki, H., & Matsubayashi, T. (2020). Mental Health Status of the General Population in Japan during the COVID-19 Pandemic. *Psychiatry and Clinical Neurosciences*. Advance online publication. <https://doi.org/10.1111/pcn.13105>

Methods

- Please include more information on the recruitment process. How did you approach them? Was there any form of compensation? Was data assessment anonymous?
- What kind of online survey tool did you use?
- P8, L40: How did you modify the instruments to „phrase the questions specific to Covid-19“. Did you modify the K6 and IESR? If you it would be good to state this more clear and describe how exactly this was done.
- Following what strategy did you enter the variables to the regression model?
- Was there missing data? If yes, how did you handle it? (This is reported in individual analysis but I think an additional general missing data statement would be of help)
- P12,L20: „Among trainees without active clinical duty during the outbreak, individuals who showed significant distress were most concerned of mental health (OR 2.41, 95% CI 1.91-3.05; Table 4) and strongly demanded personal protective equipment.....“. I think this should interpreted more carefully as this hints at a causal association. It could be also the other way round that in places where problems with protective equipment exist (might be different even in the same hospital), mental burden is higher

Results:

- I would recommend to report exact p values (and only for values smaller than .01 p<.01)
- Table 2: Are these all results from multiple regression? I would name the analysis you did in the header or legend. I am a bit confused with the „Mean“ colum. What exactly does it show? If it means why are they all ranging around 0? Did you transform the variables? Or are it the intercepts of the regression model? But where do the SDs come from then? You see, I am a bit confused. I think it would be important to make this clear for the reader.
- I think it would be interesting to show means and SDs of all outcome variables, ideally with CIs for all relevant outcome means
- Table 4. Several factors are shown in relation to psychological distress and ASR. The proportions are shown regarding the

	reference category (e.g. psychological distress). I wonder it might be more interesting for the reader if the prevalence of e.g. psychological distress is shown regarding the factor. E.g. instead of „Of the people with severe distress, how many are infected with Coronavirus“ -> „Of the people infected with Coronavirus, how many people show severe distress“. In general this is mathematically basically the same but the other way around it would be more interesting? What do you think? - CIs are reported in some places separated by , and in some separated by – I think it would be good to unify this Discussion: - P15,L35: „First, the response rate was low in residents and the non-participants were more likely to be the ones with severe stress“ This sound like this would have been tested somehow. I would rephrase to make clear that this is a hypothesis. - I think it would be important to state clearly in the limitations that the generalizability to other hospitals and medical populations remains unclear (you state it indirect but for unexperienced readers I think it would be important to say it more clear) - I would also state in the limitations that there was no baseline before the pandemic and one does not know how high levels of stress and ASR are in general in this population
--	---

REVIEWER	Carolina S Romero García Consortio Hospital General Universitario de Valencia
REVIEW RETURNED	16-Jul-2020

GENERAL COMMENTS	Good article of an interesting and important topic. 1.- TABLE 2 and TABLE 3: the results presented are not clear where they come from, ex table 3 Medicine, no duty -0.00 ± 1.02. 2.- An appendix with all the questions that were asked in the survey could clarify better the topic.
--

VERSION 1 – AUTHOR RESPONSE

Reviewer: 1

Reviewer Name: Moritz Bruno Petzold

Institution and Country: Charité Universitätsmedizin Berlin, Department for Psychiatry and Psychotherapy

Charité Campus Mitte, Berlin, Germany

Please state any competing interests or state 'None declared': None declared

Please leave your comments for the authors below

Thank you very much for the opportunity to review this manuscript. The authors present a cross-sectional examination of psychological burden in medical trainees in a single medical hospital in Sichuan, China. The manuscript addresses a very important topic, where substantial research is urgently needed. The paper is in large parts well written, concise, informative and substantial. The authors recruited a fairly large sample. Despite some limitations (single hospital, cross-sectional), the paper does come with substantial information and does contribute something to this important and timely area of research.

Nevertheless, there are some points which would have to be addressed before the manuscript could be published:

Authors' response: Thanks for the positive comments. We have now addressed the concerns point-by-point as below.

General points:

- The manuscript needs a carefully re-read to correct some typographical and grammatical errors. E.g. Abstract: „To assess psychological distress among healthcare trainees during the COVID-19 outbreak are necessary, P10,L38: „liner“, P11,L12: „then“ etc.

Authors' response: We are sorry for the typographical and grammatical errors. We have checked the manuscript carefully and done a thorough language editing.

'Abstract' section (page 3, line 4-5):

“It is essential to assess the psychological distress experienced by healthcare trainees during the COVID-19 outbreak.”

'Statistical Analysis' section (page 11, line 3):

“Differences in symptom scores or the probability of cases were estimated using linear regression (β coefficients) and logistic regression (odds ratios, ORs), respectively.”

'Result' section (page 12, line 2):

“Approximately one-fifth (19.79%) of participants were involved in active clinical duties; among them, 74 (8.94%) were working on the front lines and 38 (4.59%) had direct contact with patients with COVID-19 (Table 1).”

- You are changing between two and three decimal places, I think it would be good to unify this. I think two decimal places are sufficient.

Authors' response: Thanks for pointing this out. We have now used two decimal places throughout the manuscript.

For instance, in the 'Method' section (page 9, line 10, 19), we wrote:

“Psychological distress was assessed The Cronbach's alpha was 0.91 in our study, indicating good scale reliability.” and “Acute stress reaction was evaluated The Cronbach's α was 0.91 in our study, suggesting good scale reliability.”

Abstract:

- P3,L42: „No evident increase was found in trainees of medical technology.“ The use of the word increase suggests that a development over time would have been assessed, which was not the case. I would suggest to change wording here.

Authors' response: We have rephrased the sentence as below to avoid such confusion.

'Abstract' section (page 3, line 17-18):

“Compared to the nursing trainees, the medical trainees (OR 1.54, 95% CI 1.22 - 1.95) reported a higher burden of psychological distress during the outbreak, whilst the medical-technology trainees (OR 1.25, 95%CI 0.97 - 1.62) reported similar symptom scores.”

- P3, L47: „(ORs 1.62-1.66)“ Is this the confidence interval? The OR estimate itself seems missing (compare with the next bracket where you report a OR followed by the corresponding CI)

Authors' response: We have now presented the results (has been updated due to new adjustment strategy) as ORs with corresponding 95% CIs.

'Abstract' section (page 3, line 18):

“Postgraduates (OR 1.55, 95%CI 1.16 - 2.08) in medicine had higher levels of distress than their

undergraduate counterparts did.”

- P3,L47: „whereas a lower burden endorsed by nursing residents (OR 0.35, 95% CI 0.19-0.63)“ Compared to whom? The beginning of the sentence says compared to undergraduate students”. I think this can't be correct as this are two different levels of comparison (training stage vs. training program)

Authors' response: Thank you. Here, we did a comparison of different training stages within a specific training program. To clarify this, we rephrased the sentence and please refer to the sentence in the revised manuscript.

'Abstract' section (page 3, line 19-20):

“Postgraduates (OR 1.55, 95%CI 1.16 - 2.08) in medicine had higher levels of distress than their undergraduate counterparts did, whereas the nursing residents (OR 0.38, 95%CI 0.20 - 0.71) reported a lower burden than did nursing undergraduates.”

Strengths and limitations:

- P5,L6: I would suggest to erase the word “comprehensively”. In my opinion you used well suited and reliable instruments to assess psychological distress, nevertheless, psychological distress is a very broad concept and a “comprehensive” assessment would have to include a more detailed assessment in my opinion.

Authors' response: Thank you. We have now deleted the word, as request.

Introduction:

- The introduction is very short. I think it would be necessary to introduce more literature on specific stressors in healthcares and to refer to the multiple commentaries and studies that got published in the last weeks. A short review of the current body of research, what kind of studies exist, what are the first insights and what are methodological problems (in brief) would be important in my opinion. To help you with that here some papers that might be suited for that. This is meant as help and I don't consider it necessary that you cite all and exactly these papers.

o Daugherty, A. M., & Arble, E. P. (2020). Prevalence of mental health symptoms in residential healthcare workers in Michigan during the covid-19 pandemic. *Psychiatry Research*, 291, 113266. <https://doi.org/10.1016/j.psychres.2020.113266>

o Chew, N. W. S., Lee, G. K. H., Tan, B. Y. Q., Jing, M., Goh, Y., Ngiam, N. J. H., . . . Sharma, V. K. (2020). A multinational, multicentre study on the psychological outcomes and associated physical symptoms amongst healthcare workers during COVID-19 outbreak. *Brain, Behavior, and Immunity*. Advance online publication. <https://doi.org/10.1016/j.bbi.2020.04.049>

o Naser, A. Y., Dahmash, E. Z., Al-Rousan, R., Alwafi, H., Alrawashdeh, H. M., Ghoul, I., . . . Alyami, H. S. (2020). Mental health status of the general population, healthcare professionals, and university students during 2019 coronavirus disease outbreak in Jordan: A cross-sectional study. *Brain and Behavior*, e01730. <https://doi.org/10.1002/brb3.1730>

o Shechter, A., Diaz, F., Moise, N., Anstey, D. E., Ye, S., Agarwal, S., . . . Abdalla, M. (2020). Psychological distress, coping behaviors, and preferences for support among New York healthcare workers during the COVID-19 pandemic. *General Hospital Psychiatry*, 66, 1–8. <https://doi.org/10.1016/j.genhosppsych.2020.06.007>

o Tan, B. Y. Q., Chew, N. W. S., Lee, G. K. H., Jing, M., Goh, Y., Yeo, L. L. L., . . . Sharma, V. K. (2020). Psychological Impact of the COVID-19 Pandemic on Health Care Workers in Singapore. *Annals of Internal Medicine*. Advance online publication. <https://doi.org/10.7326/M20-1083>

o Zerbini, G., Ebigbo, A., Reicherts, P., Kunz, M., & Messman, H. (2020). Psychosocial burden of healthcare professionals in times of COVID-19 - a survey conducted at the University Hospital Augsburg. *German Medical Science : GMS E-Journal*, 18, Doc05. <https://doi.org/10.3205/000281>

o Petzold, M. B., Plag, J., & Ströhle, A. (2020). Dealing with psychological distress by healthcare

professionals during the COVID-19 pandemic [Dealing with psychological distress by healthcare professionals during the COVID-19 pandemic]. *Der Nervenarzt*, 91(5), 417–421.
<https://doi.org/10.1007/s00115-020-00905-0>

Authors' response: Thank you so much for the suggestions and sharing the citations. We have now extended the Introduction section.

'Introduction' section (page 6 line 3-12; 20-22):

"The ongoing global pandemic of the 2019 novel coronavirus disease (COVID-19) has caused 1,991,562 cases and 130,885 deaths as of April 16th, 2020(1). Witnessing an unexpected illness or death, fear of being in direct contact with and infected by patients with COVID-19, and dealing with household financial hardships during the outbreak has increased the mental burden in the general population(2). These factors have also elevated the mental burden of healthcare trainees and workers(3-5), with frontline workers having heavy workloads and being placed at higher risk for COVID-19, due to the drastic surge in patients with COVID-19. Emerging data indicate that Chinese healthcare workers exposed to COVID-19 have experienced psychological symptoms, especially women, nurses, those in Wuhan (the first epicenter), and frontline workers(6). Other studies have reported a profound mental impact of the COVID-19 outbreak on healthcare workers globally(3, 5, 7).

Despite their limited direct contact with patients with COVID-19, healthcare trainees are a vulnerable group(8). As the pandemic escalates, many countries are considering, or have already graduated senior students earlier to assist frontline workers. Other aggressive approaches have been proposed, for instance, suspending all medical school education for one year and recruiting medical students for testing, tracking, and quarantining patients with COVID-19(9). Although many trainees are inspired during these unprecedented times, some, especially those without sufficient clinical experience, may experience stress. Nevertheless, the psychological state of healthcare trainees across various programs and training stages, in response to the COVID-19 outbreak, is unknown."

- Furthermore, some introducing sentences on the impact of the pandemic on the general population and that healthcare workers of course also suffer from the general factors (and in addition to specific factors related to their profession) would be interesting for the reader, I think. Again, some help with some papers which might be suitable (but again: I do not expect to cite all of them):

o Le Shi, Lu, Z.-A., Que, J.-Y., Huang, X.-L., Liu, L., Ran, M.-S., . . . Lu, L. (2020). Prevalence of and Risk Factors Associated with Mental Health Symptoms Among the General Population in China During the Coronavirus Disease 2019 Pandemic. *JAMA Network Open*, 3(7), e2014053.

<https://doi.org/10.1001/jamanetworkopen.2020.14053>

o Petzold, M. B., Bendau, A., Plag, J., Pyrkosch, L., Mascarell Maricic, L., Betzler, F., . . . Ströhle, A. (2020). Risk, resilience, psychological distress, and anxiety at the beginning of the COVID-19 pandemic in Germany. *Brain and Behavior*, e01745. <https://doi.org/10.1002/brb3.1745>

o Ueda, M., Stickley, A., Sueki, H., & Matsubayashi, T. (2020). Mental Health Status of the General Population in Japan during the COVID-19 Pandemic. *Psychiatry and Clinical Neurosciences*. Advance online publication. <https://doi.org/10.1111/pcn.13105>

Authors' response: Thanks for your comments. We fully agree with the reviewer that healthcare workers can also suffer from the general factors, and we now have added some relevant statements in the revised manuscript

'Introduction' section (page 6, line 3-8):

"The ongoing global pandemic of the 2019 novel coronavirus disease (COVID-19) has caused 1,991,562 cases and 130,885 deaths as of April 16th, 2020(1). Witnessing an unexpected illness or death, fear of being in direct contact with and infected by patients with COVID-19, and dealing with household financial hardships during the outbreak has increased the mental burden in the general

population(2). These factors have also elevated the mental burden of healthcare trainees and workers(3-5), with frontline workers having heavy workloads and being placed at higher risk for COVID-19, due to the drastic surge in patients with COVID-19. Emerging data indicate that Chinese healthcare workers exposed to COVID-19 have experienced psychological symptoms, especially women, nurses, those in Wuhan (the first epicenter), and frontline workers(6). Other studies have reported a profound mental impact of the COVID-19 outbreak on healthcare workers globally(3, 5, 7).”

Methods

- Please include more information on the recruitment process. How did you approach them? Was there any form of compensation? Was data assessment anonymous?

Author’s response: Thank you. We add more information on the recruitment process in Study design section. In the present survey, answers to these electronic questionnaires were collected anonymously. And no compensation was offered for participants.

Study design (page 7, line 2-9):

“We conducted a cross-sectional study of healthcare trainees from the West China School of Medicine and West China Hospital, Sichuan University during February 7th-13th, 2020. We invited 7177 individuals, including 2483 undergraduates, 2606 postgraduates, and 2088 residents, to participate in this study to assess their mental health and working conditions during the COVID-19 outbreak via WeChat, a popular social media application in China. The 4184 trainees who agreed to participate were included in the analyses. For data protection, answers to these electronic questionnaires were kept anonymously. The response rates for undergraduates, postgraduates, and residents were 73.22%, 71.49%, and 24.09%, respectively (Supplementary Figure 1).”

- What kind of online survey tool did you use?

Authors’ response: Thank you. To clarify this, we rephrased the sentence and please refer to the statement in the revised manuscript.

‘Method’ section (page 7, line 5):

“We conducted a cross-sectional study of healthcare trainees from the West China School of Medicine and West China Hospital, Sichuan University during February 7th-13th, 2020. We invited 7177 individuals, including 2483 undergraduates, 2606 postgraduates, and 2088 residents, to participate in this study to assess their mental health and working conditions during the COVID-19 outbreak via WeChat, a popular social media application in China.”

- P8, L40: How did you modify the instruments to „phrase the questions specific to Covid-19“. Did you modify the K6 and IESR? If you it would be good to state this clearer and describe how exactly this was done.

Authors’ response: To clarify the modifications, we have now provided the questionnaires we used in the appendix.

‘Supplemental Text’ Section (Supplementary page 3-4):

“Psychological distress

Since the COVID-19 outbreak,

a) How often did you feel nervous?

b) How often did you feel hopeless?

c) How often did you feel restless or fidgety?

d) How often did you feel so depressed that nothing could cheer you up?

e) How often did you feel that everything was an effort?

f) How often did you feel worthless?”

“Acute stress reaction

During the past seven days concerning the COVID-19 outbreak, how much have you been distressed or bothered by these difficulties?

- a) Any reminder brought back feelings about it.
- b) I had trouble staying asleep.
- c) Other things kept making me think about it.
- d) I felt irritable and angry.
- e) I avoided letting myself get upset when I thought about it or reminded me of it.
- f) I thought about it when I did not mean to.
- g) I felt as if it had not happened or was not real.
- h) I stayed away from reminders of it.
- i) Pictures about it popped into my mind.
- j) I was jumpy and easily startled.
- k) I tried not to think about it.
- l) I was aware that I still had many feelings about it, but I did not deal with them.
- m) My feelings about it were kind of numb.
- n) I found myself acting or feeling like I was back at that time.
- o) I had trouble falling asleep.
- p) I had waves of strong feelings about it.
- q) I tried to remove it from my memory.
- r) I had trouble concentrating.
- s) Reminders of it caused me to have physical reactions, such as sweating, trouble breathing, nausea, or a pounding heart.
- t) I had dreams about it.
- u) I felt watchful and on-guard.
- v) I tried not to talk about it.”

-Following what strategy did you enter the variables to the regression model?

Authors' response: Thank you for your important comment. We previously only adjusted for age and sex in the original manuscript. For having a better control for the potential underlying confounders, in the revised manuscript, we included more variables in the regression models including well-established risk factors of psychological distress according to literature, as well as variables that unevenly distributed in our studied groups. Given the relatively large sample size of our study, we included all relevant covariates in the model, without any selection.

We reported our strategy in the ‘Statistical Analysis’ section and below Table 2-4 (page 22-26).

Specifically, previous studies reported that sex, age, marriage, epidemic contact characteristics were the influence factors of psychological distress (Castaldelli-Maia JM, *Int Rev Psychiatry*, 2019; Huiyao Wang, *PLoS One*, 2020). Additionally, in our study, Table 1 indicated the significantly different distributions of characteristics including age, sex (male or female), marital status (unmarried or married), training stage (undergraduate, postgraduate or residency), location (Hubei or outside Hubei), relatives with COVID-19 (yes or no), active clinical duty (yes or no), working position (frontline or second-line), contact with COVID-19 (yes or no), among studied groups. We therefore included them (if applicable) in the corresponding models. However, as the status of clinical duty is strongly correlated with training stage (few undergraduates and postgraduates were with active clinical duty but all the residency had active clinical duty), we only kept the training stage in the model.

‘Statistical Analysis’ section (page 11, line 7-12):

“All models were adjusted for age, sex, marital status and epidemic contact characteristics to address confounding by these variables. We also adjusted the model for training program and training stage when analyzing the associations of concerns, needs, career impact and family-work conflicts with psychological distress and ASR. As the status of clinical duty is strongly correlated with training stage, we didn't adjust for active clinical duty (yes or no) as covariates.”

In brief, we obtained largely similar estimates after the change of adjustment strategy. Correspondingly, we have updated all relevant contents in the revised manuscript.

- Was there missing data? If yes, how did you handle it? (This is reported in individual analysis but I think an additional general missing data statement would be of help)

Authors' response: Thanks for your comments. Given the data was collected by the electronic questionnaires, we don't have missing data for general conditions. However, not all participants returned complete forms of psychological scales 462 (11.04%) missed the measure of psychological distress, and 433 (10.35%) missed the scores of acute stress reaction. They were therefore not included in the corresponding analysis. The rate of missing data was reported below table 2-5 (page 22-26). We have added an additional general missing data statement as below the 'Statistical Analysis' section.

Statistical Analysis (page 11, line 12-14):

"Individuals with missing data on the measures of psychological distress (462, 11.04%) or ASR (433, 10.35%) were not included in the corresponding analyses."

Results:

- P12,L20: „Among trainees without active clinical duty during the outbreak, individuals who showed significant distress were most concerned of mental health (OR 2.41, 95% CI 1.91-3.05; Table 4) and strongly demanded personal protective equipment.....“. I think this should be interpreted more carefully as this hints at a causal association. It could be also the other way round that in places where problems with protective equipment exist (might be different even in the same hospital), mental burden is higher

Authors' response: We fully agree with the reviewer. We have now rephrased the sentence and please refer to the sentence in the revised manuscript. And the statistics changed because we had revised covariates of regression models according to the comment of the first reviewer.

'Results' section (page 13, line 11-13):

"Among the trainees without active clinical duties during the outbreak, psychological distress was significantly associated with concerns about mental health (OR 2.41, 95%CI 1.90 - 3.04; Table 4) and demands for personal protective equipment (OR 1.51, 95%CI 1.07 - 2.16)."

- I would recommend to report exact p values (and only for values smaller than .01 $p < .01$)

Authors' response: We have now reported exact p values as suggested. Please refer to the sentence in the revised manuscript.

'Results' section (page 12, line 5-8).

"Compared with trainees in the medical technology and nursing programs, the trainees in medicine were older ($p < 0.01$), and more likely to be postgraduates ($p < 0.01$), males ($p < 0.01$), married ($p < 0.01$), and living outside of Hubei Province, the first epicenter ($p = 0.02$). At the time of the survey, nursing trainees were more likely to have active clinical duties and work on the front lines ($p < 0.01$)."

- Table 2: Are these all results from multiple regression? I would name the analysis you did in the header or legend. I am a bit confused with the „Mean“ column. What exactly does it show? If it means why are they all ranging around 0? Did you transform the variables? Or are they the intercepts of the regression model? But where do the SDs come from then? You see, I am a bit confused. I think it would be important to make this clear for the reader.

Authors' Response: Thanks for pointing this out. We converted the total score of psychological distress and acute stress response assessment to z-score for analysis. We have clarified this in the Method and Tables. Please refer to the sentence in the revised manuscript,

'Statistical Analysis' section (page 10, line 24-25; page 11, line 1-4):

“We described the distributions of the symptoms’ scores (transformed z-scores are reported as mean standard deviation), and the proportion of identified cases (corresponding to the cut-off points stated in the Methods section), in each of the three program groups. Differences in symptom scores or the probability of cases were estimated using linear regression (β coefficients) and logistic regression (odds ratios, ORs), respectively.”

- I think it would be interesting to show means and SDs of all outcome variables, ideally with CIs for all relevant outcome means

Authors’ response: We agree with the reviewer that we should report means and SDs for the relevant outcome. We have now added them in Table 2. Please refer to the sentence in the revised manuscript, ‘Results’ section Table 2 (page 23). We are willing to provide more information if need.

- Table 4. Several factors are shown in relation to psychological distress and ASR. The proportions are shown regarding the reference category (e.g. psychological distress). I wonder it might be more interesting for the reader if the prevalence of e.g. psychological distress is shown regarding the factor. E.g. instead of „Of the people with severe distress, how many are infected with Coronavirus“ -> „Of the people infected with coronavirus, how many people show severe distress“. In general, this is mathematically basically the same but the other way around it would be more interesting? What do you think?

Authors’ response: Thank you for these important comments. We fully agree that it will be more appropriate to report the proportion of each concern/need/future career choice by with and without psychological distress/acute stress reaction. We have recalculated and revised Table 4 accordingly (page 25).

- CIs are reported in some places separated by , and in some separated by – I think it would be good to unify this.

Authors’ response: Thanks for catching it. We have now unified the presentation by using “-” in between. Please refer to the sentence in the revised manuscript, ‘Table 2-5’ section (page 23-27).

Discussion:

- P15,L35: „First, the response rate was low in residents and the non-participants were more likely to be the ones with severe stress“ This sound like this would have been tested somehow. I would rephrase to make clear that this is a hypothesis.

Authors’ response: Thank you for these comments. We fully agree with you, and we have clarified in the ‘limitation’ section.

‘limitation’ section (page 16, line 13-17):

“Second, the response rate was low among the residents, and those who did not participate might have been the trainees with highest stress levels at work. The participation rates of undergraduate and postgraduate trainees were satisfactory (73.22% and 71.49% respectively). Such selection is not likely to provide a thorough explanation of our findings.”

- I think it would be important to state clearly in the limitations that the generalizability to other hospitals and medical populations remains unclear (you state it indirect but for unexperienced readers I think it would be important to say it clearer)

Authors’ response: We agree with the reviewer and have clarified this in the revised version ‘limitation’ section.

‘limitation’ section (page16, line 24-25):

“Last, although the trainees came from all parts of China, our study was conducted at a single medical

school and teaching hospital. The generalizability of our findings to other hospitals and medical populations remains unclear, and therefore, needs further investigation.”

- I would also state in the limitations that there was no baseline before the pandemic and one does not know how high levels of stress and ASR are in general in this population

Authors’ response: This is an important point. We have clarified it in the revised version ‘limitation’ section.

‘limitation’ section (page 16, line 11-13):

“First, given the nature of cross-sectional analyses, our data do not indicate changes in psychological distress from the pre-pandemic period; rather, they characterize the burden during the COVID-19 outbreak.”

Reviewer: 2

Reviewer Name: Carolina S Romero García

Institution and Country: Consorcio Hospital General Universitario de Valencia

Please state any competing interests or state ‘None declared’: none declared

Please leave your comments for the authors below

Good article of an interesting and important topic.

1.- TABLE 2 and TABLE 3: the results presented are not clear where they come from, ex table 3 Medicine, no duty -0.00 ± 1.02 .

Authors’ response: Thank you for your comments. For facilitate the analyses, we have converted the total score of psychological distress and acute stress response assessment to z-score. According to the journal style, results are reported as two decimals. Therefore, statistics less than 0.01 were shown as 0.00. In revised manuscript, we clarified these details and we also add notes, for explanations, in Table 3.

‘Statistical analysis’ section (page 10, line 24-25; page 11, line 1-4):

“We described the distributions of the symptoms’ scores (transformed z-scores are reported as mean standard deviation), and the proportion of identified cases (corresponding to the cut-off points stated in the Methods section), in each of the three program groups. Differences in symptom scores or the probability of cases were estimated using linear regression (β coefficients) and logistic regression (odds ratios, ORs), respectively.”

‘Tables 3’ section (page 24, line 5):

“d. $-0.00 < -0.01$ ”

2.- An appendix with all the questions that were asked in the survey could clarify better the topic.

Authors’ response: Thanks for the suggestion. We have now provided all modified questionnaires in the supplement (see below).

‘Supplemental Text’ Section (Supplementary page 2-4):

“Demographic information

a) Age: _____ years

b) Sex: A male, B female

c) Training stage: A undergraduate, B postgraduate, C residency

d) Training program: A medicine, B medical technology, C nursing

e) Marital status: A married, B unmarried

f) Location: _____ (current)

g) Have you had relatives infected with COVID-19?

A. No B. Yes

h) Are you active in clinical duty?
A. No B. Yes

If Yes, go on:

- i) Working position: _____ department.
- ii) Since the outbreak, have you contacted with COVID-19 patients?
A. Yes B. No
- iii) Work-family conflict and support
Since the outbreak,
 - a) my current job has made it difficult for me to care for my family.
 - b) family responsibilities have affected my work.
 - c) I had difficulties in juggling work and family.
 - d) I can get support from my family.
 - e) I can get support from my colleagues.
 - f) I can get support from my leader.
 - g) The hospital's holiday arrangement is reasonable.
 - h) The hospital's duty arrangement is reasonable.
 - i) The hospital has had a flexible policy that allowed me to juggle family and work.

If No, go on:

- i) Concerns during the outbreak (Multiple choices)
Under the current circumstances, I am concerned about
 - a) being infected with novel coronavirus;
 - b) my physical health condition;
 - c) my psychological health;
 - d) academic performance;
 - e) my social life/work;
 - f) my traveling plan;
 - g) the risk of infection for family members or friends;
 - h) my personal or family financial situation;
 - i) other things.
- ii) Needs during the outbreak (Multiple choices)
Under the current circumstances, if I were to work during the outbreak, I need
 - a) personal protective equipment;
 - b) social insurance;
 - c) salary incentives;
 - d) clinical practice guidance;
 - e) professional track record;
 - f) others.
- iii) The influence on future career choice (single question)
Has the outbreak affected your future career plan? i) Healthcare worker; ii) Medicine-related, but not bedside; iii) Outside of medicine; iv) Indeterminate.

Psychological distress

Since the COVID-19 outbreak,

- a) How often did you feel nervous?
- b) How often did you feel hopeless?
- c) How often did you feel restless or fidgety?
- d) How often did you feel so depressed that nothing could cheer you up?

- e) How often did you feel that everything was an effort?
- f) How often did you feel worthless?

Acute stress reaction

During the past seven days concerning the COVID-19 outbreak, how much have you been distressed or bothered by these difficulties?

- a) Any reminder brought back feelings about it.
- b) I had trouble staying asleep.
- c) Other things kept making me think about it.
- d) I felt irritable and angry.
- e) I avoided letting myself get upset when I thought about it or reminded me of it.
- f) I thought about it when I did not mean to.
- g) I felt as if it had not happened or was not real.
- h) I stayed away from reminders of it.
- i) Pictures about it popped into my mind.
- j) I was jumpy and easily startled.
- k) I tried not to think about it.
- l) I was aware that I still had many feelings about it, but I did not deal with them.
- m) My feelings about it were kind of numb.
- n) I found myself acting or feeling like I was back at that time.
- o) I had trouble falling asleep.
- p) I had waves of strong feelings about it.
- q) I tried to remove it from my memory.
- r) I had trouble concentrating.
- s) Reminders of it caused me to have physical reactions, such as sweating, trouble breathing, nausea, or a pounding heart.
- t) I had dreams about it.
- u) I felt watchful and on-guard.
- v) I tried not to talk about it.”

VERSION 2 – REVIEW

REVIEWER	Dr. rer. medic. Moritz Petzold Charité Universitätsmedizin Berlin - Department for Psychiatry and Psychotherapy, Campus Charité Mitte, Berlin, Germany
REVIEW RETURNED	02-Sep-2020

GENERAL COMMENTS	The authors did a great job and addressed all the comments. The manuscript significantly improved by this.
--

REVIEWER	Carolina S Romero García Consorcio Hospital General Universitario de Valencia
REVIEW RETURNED	18-Sep-2020

GENERAL COMMENTS	The reviewer appreciates the opportunity assess this interesting work and congratulates the authors for the significant improve in the manuscript.
--